# Impact of Exosomes Released by Different Corneal Cell Types on the Wound Healing Properties of Human Corneal Epithelial Cells

**DOI:** 10.3390/ijms232012201

**Published:** 2022-10-13

**Authors:** Pascale Desjardins, Rébecca Berthiaume, Camille Couture, Gaëtan Le-Bel, Vincent Roy, François Gros-Louis, Véronique J. Moulin, Stéphanie Proulx, Sylvain Chemtob, Lucie Germain, Sylvain L. Guérin

**Affiliations:** 1Regenerative Medicine Division of the Centre de Recherche du CHU de Québec, Université Laval, Québec, QC G1J 1Z4, Canada; 2Centre Universitaire d’Ophtalmologie (CUO)-Recherche, Hôpital du Saint-Sacrement, 1050 Chemin Ste-Foy, Québec, QC G1J 1Z4, Canada; 3Centre de Recherche en Organogénèse Expérimentale de l’Université Laval/LOEX, Hôpital Enfant-Jésus, 1401 18e Rue, Québec, QC G1V 0A6, Canada; 4Département d’Ophtalmologie, Faculté de Médecine, Université Laval, Québec, QC G1V 0A6, Canada; 5Département de Chirurgie, Faculté de Médecine, Université Laval, Québec, QC G1V 0A6, Canada; 6Département d’Ophtalmologie, Faculté de Médecine, Université de Montréal, Montréal, QC H3T 1J4, Canada

**Keywords:** cornea, wound healing, exosomes, extracellular vesicles, corneal epithelial cells, corneal stromal fibroblasts, corneal endothelial cells, signal transduction pathway

## Abstract

Corneal wound healing involves communication between the different cell types that constitute the three cellular layers of the cornea (epithelium, stroma and endothelium), a process ensured in part by a category of extracellular vesicles called exosomes. In the present study, we isolated exosomes released by primary cultured human corneal epithelial cells (hCECs), corneal fibroblasts (hCFs) and corneal endothelial cells (hCEnCs) and determined whether they have wound healing characteristics of their own and to which point they modify the genetic and proteomic pattern of these cell types. Exosomes released by all three cell types significantly accelerated wound closure of scratch-wounded hCECs in vitro compared to controls (without exosomes). Profiling of activated kinases revealed that exosomes from human corneal cells caused the activation of signal transduction mediators that belong to the HSP27, STAT, β-catenin, GSK-3β and p38 pathways. Most of all, data from gene profiling analyses indicated that exosomes, irrespective of their cellular origin, alter a restricted subset of genes that are completely different between each targeted cell type (hCECs, hCFS, hCEnCs). Analysis of the genes specifically differentially regulated for a given cell-type in the microarray data using the Ingenuity Pathway Analysis (IPA) software revealed that the mean gene expression profile of hCECs cultured in the presence of exosomes would likely promote cell proliferation and migration whereas it would reduce differentiation when compared to control cells. Collectively, our findings represent a conceptual advance in understanding the mechanisms of corneal wound repair that may ultimately open new avenues for the development of novel therapeutic approaches to improve closure of corneal wounds.

## 1. Introduction

The eye is a fascinating, yet complex organ composed of many structures that allows us to see our surrounding. One such structure is the cornea, a particularly important tissue as its structural integrity is crucial for proper light transmission to the retina. In fact, corneal transparency is critical to ensure a proper visual acuity. However, because of its anterior position at the surface of the eye, the cornea is subjected to traumas such as chemical and mechanical injuries that may disturb the vision. Abrasions of the corneal epithelium caused by fingernails [1] or sustained contact lenses wear [2] account for a large proportion of the corneal injuries. If not treated properly or rapidly, they can lead to infections, which may then progress toward more serious complications, such as ulcers and corneal perforations [3]. Moreover, traumatic events such as the propulsion of metallic debris [4,5] or chemical substances [6] in the eye can deteriorate the eye surface at a point where corneal stem cells from the limbus can no longer ensure regeneration of the corneal tissue. This can lead to corneal opacification and in more severe cases, to a complete loss of vision as a result of a syndrome known as the limbal stem cell deficiency (LSCD) [7]. In every case, a rapid and appropriate treatment of the injury allows a better recovery of visual acuity [8,9].

Corneal wound healing requires many biological processes such as cell proliferation, migration, adhesion, differentiation as well as extracellular matrix (ECM) remodeling [10]. One key aspect that regulates these events resides in the communication between cells present in each of the three corneal layers (epithelium, stroma and endothelium). The epithelium, the most outer layer of the cornea, consists of five to seven well-organized layers of corneal epithelial cells. The stroma is the thicker segment of the cornea. It is mainly composed of collagen types I and V organized into collagen fibrils, both components being secreted by the stromal fibroblasts. In turn, collagen fibrils are then organized into flattened lamellae, which are precisely superimposed one on another in a manner that allows light transmission to the retina [11]. A monolayer of endothelial cells lines the posterior limit of the cornea and actively regulates stromal hydration. Much evidence suggests that the cells from these three different corneal compartments communicate with each other to facilitate wound healing upon injury, especially when the basement membrane (BM) underneath the epithelium is disrupted. For example, abrasions that affect the epithelium prompt the secretion of fibronectin by epithelial cells as well as stromal fibroblasts. Such newly formed ECM then facilitates the migration of the epithelial cells to cover the wound [12]. In addition, it is known that cytokines such as TGF-β1 and PDGF secreted by epithelial cells can pass through the discontinued BM in order to trigger differentiation of the stromal fibroblasts into myofibroblasts, which is useful for the wounding process [13]. Moreover, these myofibroblasts produce growth factors such as HGF and KGF, which will in turn contribute to the proliferation of the corneal epithelial cells [14]. The endothelial-mesenchymal transition also represents another such example, this process being triggered by the secretion of FGF-2 by corneal epithelial cells during epithelial wound healing [15,16].

Despite the fact that there is a clear communication between the three layers of the cornea, little is known about how this communication occurs. This process undoubtedly involves extracellular vesicles (EVs), and more particularly a relatively small class of EVs called exosomes. Exosomes typically range from 30 to 150 nm of diameter and are derived from multivesicular bodies that escape the lysosomal degradation [17,18,19]. When released by the cell through exocytosis [20], exosomes can either act on nearby cells or travel long distances via the blood and lymphatic vessels [21,22] to reach other tissues or organs. An interesting feature of the exosomes is their capacity to carry various bioactive cargos such as lipids [23], proteins [24,25], miRNAs [26,27,28,29], long non-coding RNAs [30,31], etc. By taking part into intercellular communication, exosomes are indeed involved in maintaining biological homeostasis, as well as in a plethora of other biological and pathological processes, including inflammation, cancer progression and wound healing [32,33,34,35]. Furthermore, because they can carry various molecules, exosomes are now being tested as drug carriers to treat diseases such as Parkinson disease [36], pancreatic cancers [37] and lung diseases [38]. The use of exosomes as low-invasive biomarkers for detecting and monitoring several pathologies is also an active area of investigation [39].

Although exosomes were discovered in the early nineties [40], their study gained popularity only 30 years later. This explains why little is known about exosomes, especially in the cornea. However, it was established that corneal epithelial cells [41], stromal fibroblasts [42] and endothelial cells [43] indeed release exosomes [44]. Moreover, EVs have been found to be secreted in the cornea following wounding [41] and it is known that exosomes secreted by corneal epithelial cells can pass through the BM if impaired by an injury [41]. These results all pointed to a potential role of exosomes in corneal wound healing, which has recently been the focus of a few studies [41,42,45,46,47,48]. However, characteristics, functional roles and mechanisms of action of corneal exosomes remain to be fully investigated. In the present study, we evaluated the capacity of exosomes released by human corneal epithelial cells (hCECs), human corneal fibroblasts (hCFs) and human corneal endothelial cells (hCEnCs) to alter the genetic and proteomic profiling of each of these cell types and determined whether they have distinctive wound healing characteristics of their own.

## 2. Results

### 2.1. Characterization of the Exosomes Derived from the Three Different Corneal Cell Types

The production of exosomes has been reported to vary greatly between different cell types, especially in cells from the immune system [49]. However, and to our knowledge, no such analysis has ever been reported for the different cell types that constitute the human cornea. In order to verify the capacity of the different corneal cells to release exosomes in their surroundings, we primary cultured human corneal epithelial (hCECs), fibroblast (hCFs) and endothelial (hCEnCs) cells as monolayers. Exosomes were then enriched by ultracentrifugation from the conditioned media of these cultured cells. Figure 1A shows typical exosomes isolated from each cell population, as seen by transmission electron microscopy (TEM). Their round morphology (often with a central depression) and sizes are well-established features of such small extracellular vesicles [50]. Their size was further determined by Dynamic Light Scattering (DLS) and are shown on Figure 1B. Results indicate that exosomes’ diameters vary from 67 to 135 nm between the three corneal cell types, which correspond to the typical range of sizes for exosomes as observed in the literature [51,52]. Moreover, we also assessed the presence of the exosome markers CD9, CD63 and CD81 in our samples. Western blot analyses revealed that those isolated from hCECs and hCFs were all enriched with the typical exosomes tetraspanin protein markers CD9, CD63 and CD81 [53] whereas those from hCEnCs were positive for both CD9 and CD63 but not for CD81 (Figure 1C).

It is now widely accepted that obtaining a pure sample, exclusively enriched in exosomes, is more a utopia than a tangible prospect [54]. Although probably mostly composed of exosomes, our samples also present a little heterogeneity, as revealed by high-sensitivity, flow cytometry experiments (Appendix A). Several observations can be drawn from these analyses. First, we can see that most of the extracellular vesicles (EVs) contained in our samples present sizes ranging from 100 to 500 nm. However, a minority of them have sizes greater than 500 nm and up to 1000 nm for the densest samples. Another interesting piece of information is that different populations of extracellular vesicles, that either stain positive or negative for one or more EVs markers (AnnexinV, CD63 and CellTracker Deep Red) could be observed. This serves as a good example to illustrate heterogeneity of the EVs samples.

### 2.2. Human Corneal Exosomes Are Taken Up by hCECs and hCFs and Alter Their Growth Properties

We next investigated the uptake of exosomes enriched from hCECs by both hCECs and hCFs grown as monolayers. After incubation of cultured cells with DiI-labeled exosomes for 24 h, red fluorescent particles were observed throughout the cells’ cytoplasm of both hCECs and hCFs (Figure 2). Cells staining with Alexa Fluor 488-conjugated phalloidin which selectively label F-actin revealed that exosomes appear to be located throughout the cytoplasm and occasionally concentrated around the nuclei. We next monitored expression of the proliferation marker Ki-67 in hCECs grown for a period of 48 h with or without addition of exosomes (Figure 3). The addition of exosomes from hCECs or hCFs significantly increased the proportion of Ki-67 positive cells to 33% and 40%, respectively compared to 10% in control hCECs. In contrast, only a slight increase in the percentage of Ki-67 positive cells (14%) was observed with exosomes from hCEnCs. We also monitored expression of the proliferation marker Ki-67, but in hCFs. The hCFs are known to be highly proliferative cells, which results in a high proportion of Ki-67 positive cells for control hCFs (around 78%). Addition of exosomes, from any corneal cell types did not significantly alter the already high percentage of Ki-67 positive cells in hCFs (Appendix A).

### 2.3. Human Corneal Exosomes Accelerate Wound Closure of hCECs In Vitro

As exosomes from mesenchymal stem cells have been shown to significantly contribute to wound healing [42,55], we then wished to verify whether exosomes enriched from hCEnCs, hCECs and hCFs would similarly impact wound closure of hCECs in scratch wound assays. hCECs were selected for this experiment as most of the corneal wounds primarily involve the corneal epithelium [10,56]. Exosomes enriched from hCEnCs, hCECs and hCFs were therefore added individually to the culture media of scratch-wounded, confluent hCECs. Fresh media and exosomes were added every 48 h until the wound was completely closed for at least one condition. As shown on Figure 4A, wounds closed faster when exosomes (irrespective of their cells of origin) were added relative to control wounds that received no exosomes. Analysis of the percentage of wound area remaining over time revealed that addition of exosomes from hCECs, hCFs and hCEnCs reduced the wound surfaces at 96 h to 4.1%, 7.6% and 10.7%, respectively, and reached complete wound closure after 120 h. In contrast, for the controls, the wound surface was 22.8% at 96 h, 7% remained at 120 h and an additional 24 h incubation (to 144 h) was necessary for complete wound closure (Figure 4B). Therefore, the addition of corneal cells’ exosomes accelerates scratch wound healing of hCECs monolayers.

### 2.4. Signal Transduction Pathways Is Modified by Exosomes in hCECs

Interaction of exosomes with nearby cells has been shown to trigger the activation of intracellular signaling pathways such as JAK/STAT [57], mitogen-activated protein kinases (MAPKs) [58] or phosphatidylinositol-3OH kinase (PI3K)/Akt pathways [59]. To determine which from these exosome-activated signaling pathways contributes the most to corneal reepithelialization in our scratch-wounded model, phosphorylation levels of protein mediators from different pathways were examined using the human phospho-kinase proteome profiler array (Figure 5). Compared to controls (no added exosomes), the addition of hCECs exosomes led to a significant increase in the phosphorylation of GSK-3β, HSP27, p38α (MAPK), STAT5, STAT3 and β-*catenin* in hCECs (Figure 5B,C). The addition of hCFs exosomes activated the same mediators as those induced by hCECs exosomes except STAT3, whose activation was drastically reduced relative to control hCECs. Only HSP27 seeks its activation increased by the addition of exosomes from hCEnCs whereas that of GSK-3β, STAT5 and β-catenin was reduced. Therefore, exosomes from different corneal cells clearly have very distinctive impacts on the activation of cell signaling mediators in vitro.

### 2.5. Exosomes Modify the Gene Expression Pattern in a Cell-Specific Manner

We next exposed near confluent hCECs (Figure 6), hCFs (Appendix A) and hCEnCs (Appendix A) to exosomes isolated from all three cell-types and conducted gene profiling analyses on microarrays using total RNAs isolated from both the negative controls and from exosomes-exposed hCECs, hCFs and hCEnCs. A scatter plot analysis of the 60,000 different transcripts contained on the arrays indicated that only a limited number of genes have their level of expression modified by the addition of exosomes in hCECs, as indicated by the values of the regression curves (R^2^ = 0.9864, 0.9791 and 0.9824 for hCECs exposed to hCECs, hCFs, and hCEnCs exosomes, respectively: Figure 6A). Consistent with these results, 534, 255 and 350 genes were found to be differently regulated by more than a 2-fold factor between control hCECs and hCECs exposed to hCECs-, hCFs- or hCEnCs-derived exosomes, respectively. Interestingly, only 82 genes (which represents 15%, 32% and 23% of all the genes differentially regulated by the hCECs, hCFs and hCEnCs exosomes, respectively) were commonly modified by the three types of exosomes (Figure 6B).

Conducting a similar analysis for hCFs yielded values for the regression curves of 0.9727, 0.9801 and 0.9859 for hCFs exposed to hCFs, hCECs and hCEnCs exosomes, respectively (Appendix A). In addition, 688, 885 and 438 differently regulated genes were identified between control hCFs and hCFs exposed to hCECs-, hCFs- and hCEnCs-derived exosomes, respectively, of which 110 genes were commonly modified by the three types of exosomes (which represents 16%, 12% and 25% of all the genes differentially regulated by the hCECs, hCFs and hCEnCs exosomes, respectively) (Appendix A). Finally, hCEnCs exhibited the most perturbed transcriptomic profile upon addition of exosomes, with 1169, 544 and 1007 differentially regulated genes identified between control hCEnCs and hCEnCs exposed to hCECs-, hCFs- and hCEnCs-derived exosomes, respectively. Of all these genes, 229 were found to be modified by all types of exosomes (which represents 20%, 42% and 23% of all the genes differentially regulated by the hCECs, hCFs and hCEnCs exosomes, respectively) (Appendix A). Most interestingly, none of the genes identified as commonly modified by all three types of exosomes were common between hCECs (82 genes), hCFs (110 genes) and hCEnCs (229 genes) (Figure 6C). This result therefore clearly established the strong cell type-specific impact that corneal exosomes exert on the transcriptomic profile of corneal cells (hCECs, hCFs or hCEnCs) with which they interact.

We next examined the data files from the microarrays to sort out only the 50 genes whose expression is the most differentially regulated in hCECs that have been exposed to exosomes enriched from hCECs, hCEnCs and hCFs relative to cells unexposed to exosomes. Examination of Figure 7 indicates that 19 genes among the 50 most differentially regulated genes between controls and hCECs grown hCECs-, hCEnCs- or hCFs-exosomes are not just common to one another but also similarly modified in all conditions: the expression of 18 of these genes (indicated in red) was drastically reduced whereas that of one gene (indicated in blue) was increased. The same exercise was conducted for hCFs and hCEnCs that have been exposed to exosomes isolated from all three corneal cell-types and heatmaps of the 50 most differentially regulated genes were generated (Appendix A, respectively). Similarly, 8 and 16 genes had their expression strongly reduced in hCFs and hCEnCs, respectively (named in red on Appendix A), when they are grown in the presence of exosomes from either hCECs, hCFs or hCEnCs.

### 2.6. In Silico Prediction of Biological Functions Affected by Exosomes in hCECs and hCFs through Gene Interaction Network Analyses

We next perform a differential expression analysis on the microarray linear expression data from hCECs, hCFs and hCEnCs cultured in the presence of exosomes from either hCECs, hCEnCs or hCFs. We identified a total of 125, 333 and 390 differentially expressed genes between hCECs, hCFs and hCEnCs cultured with the three exosome types, respectively (adjusted *p*-value < 0.05 and FC ≥ 3). We then uploaded the results from these analyses into the Ingenuity Pathway Analysis (IPA) software to be further analyzed. IPA’s statistical algorithms and curated knowledge database can be used to predict what and how biological functions are likely to be influenced when provided with data from a differential expression analysis. We thus selected four biological functions of interest, “proliferation”, “differentiation”, “migration” and “immune response” of hCECs (Figure 8), hCFs (Appendix A) and hCEnCs (Appendix A), to which we connected all the differentially expressed genes that were linked to these functions according to the database. We then used IPA to examine how these genes interacted and to computationally predict how the resulting networks affected the biological functions of interest. Given our microarray data analysis, IPA predicted that hCECs cultured in the presence of exosomes from the three corneal cell types (hCECs, hCEnCs and hCFs) would proliferate and migrate more than control hCECs while, in the meantime, they would differentiate less (Figure 8). Many of the 50 most differentially expressed genes (especially in hCECs exposed to exosomes isolated from hCECs) were also recognized by IPA to impact on the different cellular processes analyzed (gene names indicated by an asterisk in Figure 7 and Appendix A). As for hCECs, analysis of the IPA data also suggest that the three types of exosomes should prompt proliferation and migration of hCFs. However, the lack of any significant difference in the expression of the proliferation marker Ki-67 (Appendix A) does not support the IPA data. In addition, the immune response is expected to be stimulated by the addition of hCECs and hCEnCs exosomes but inhibited by that of hCFs exosomes (Appendix A). The scenario became quite different in hCEnCs as the IPA analysis suggested that all three types of exosomes should interfere with endothelial cell proliferation. In addition, IPA data indicate that upon addition of exosomes, migration, differentiation and immune response in endothelial cells are going in various directions depending on the cell type from which the exosomes originate (Appendix A).

## 3. Discussion

Three major different types of extracellular vesicles have been recognized based on their respective mechanism of biogenesis: exosomes, ectosomes (including microvesicles and oncosomes, ranging in size between 100 nm to 10 µm) and apoptotic bodies (ranging in size between 1 to 5 µm). Exosomes are lipid bi-layer vesicles that originate from intracellular multivesicular bodies [38,60] and whose size vary from one study to another but are generally viewed as within 30 to 150 nm in diameter. Exosomes are very effective cell-to-cell intermediates that can deliver a whole array of compounds such as lipids, proteins and nucleic acids (including both mRNA and miRNA) [61,62,63,64,65]. They can be derived from various biological fluids such as blood, amniotic fluid, breast milk, synovial fluid, ascites and pleural effusions [66] or secreted by a large variety of cells such as fibroblasts, immune cells (both B and T cells), dendritic, neuron and intestinal cells, stem cells as well as cancer cells (reviewed in [66,67]). Exosomes also proved particularly promising at improving wound healing, especially for the skin [68,69,70]. By exploiting an epithelial debridement mouse model, Samaeekia et al. nicely demonstrated that exosomes derived from human corneal mesenchymal stromal cells could accelerate the wound closure of human corneal epithelial cells in vitro as well as in vivo [42]. Other recent studies have also highlighted the role of corneal-derived exosomes in corneal wound healing [41,71]. In the present study, we enriched exosomes from the three main cell types (epithelial, stromal and endothelial cells) of the human cornea and evaluated their capacity at modifying reepithelialization of human corneal epithelial cells in a monolayer scratch wound healing model. We demonstrated that all types of exosomes were efficiently taken up by hCECs, hCFs and hCEnCs and that they impacted on many biological functions of the recipient cells, such as proliferation and wound healing. We also found that exosomes from hCECs, hCFs and hCEnCs profoundly alter the transcriptomic profile of all three corneal cell types in a cell type-specific manner in vitro.

Characterization of EVs samples is a crucial, yet challenging step. Our results show clear size differences in the populations of exosomes between the three different types of cells, which ranged from 67 nm for exosomes enriched from hCFs, to 135 nm with those enriched from hCECs. TEM images also show EVs of different sizes ranging approximately from 50 to 150 nm. Variations in exosomes diameter is not uncommon as Rashid et al. reported they can vary from 98 to 140 nm in those released by human embryonic kidney 293 (HEK293), myeloid-derived suppressor cells (MDSCs) and endothelial progenitor cells (EPCs) [72]. In addition, the mean exosome diameter of human induced pluripotent stem cells (iPSCs) was found to be of 86 nm [73] whereas that of both adipose-derived stem cells (ADSCs) and human foreskin fibroblasts averaged 134 and 142 nm, respectively [74]. Tetraspanins is a family of transmembrane proteins that allow association with other members of the family and with other proteins to generate dynamic membrane domains. Traditionally, several members of this family, especially CD63, CD9 and CD81 were known to be highly enriched in exosomes from virtually any cell type and are commonly used as markers for exosomes identification, quantitation, or purification [75], although irregularities in their respective expression has recently been reported [76]. In our present study, all three exosome markers were found to be expressed by hCECs and hCFs exosomes but not by hCEnCs that were positive for CD63 and CD9 but not for CD81. However, and as reported by Garcia-Martin et al., the lack of CD81 expression in our endothelial exosomes is not unique as this marker was barely detectable in SVEC endothelial cells relative to CD63 [76]. However, these types of tetraspanins have also been identified in other types of EVs, which highlights the complexity of finding a unique exosomal marker [77].

One clear finding is that human corneal exosomes are taken up by hCECs and hCFs in culture. Interestingly, exosomes’ uptake appears to be more noticeable in hCECs compared to hCFs. This observation may be due to the cell distribution and level of confluence of the culture. Furthermore, the internalization process, the rate at which it occurs, and the intracellular fate of exosomes could also greatly vary depending on the cell type on which they are deposited. Once internalized, exosomes have clear effects on the biological functions of the target cells. Indeed, exosomes enriched from all types of corneal cells impacted to varying degrees on hCECs proliferation in the following order: hCFs > hCECs > hCEnCs (4-, 3.3- and 1.4-fold increase in Ki-67 positive cells, respectively). In addition, those purified from hCECs and hCFs were also those that impacted the most on wound closure of scratch-wounded hCECs. However, it is presently difficult to precisely determine the reasons why exosomes enriched from different cell types may have distinctive impacts on wound healing. We can assume that this is probably due, at least in part, to the differences inherent to each type of exosome, such as their cargo composition and the membrane receptors present at their surface. Moreover, some of our results might help understand them partly. Indeed, compared to controls (no added exosomes), addition of exosomes isolated from hCEnCs completely abolished the phosphorylation of glycogen synthase kinase 3β (GSK-3β), p38α (MAPK) and β-catenin in hCECs, whereas those isolated from hCECs and hCFs considerably increased activation of these signal transduction mediators. This is consistent with the fact that signal transduction through the GSK-3β/β-catenin pathway is abundantly documented to promote proliferation in a large variety of cell types [78,79,80,81,82]. Not surprisingly, the GSK-3β/β-catenin pathway is involved in the progression and invasiveness of many types of cancers [83,84,85,86,87]. Phosphorylation of β-catenin by GSK-3β targets β-catenin for ubiquitination and subsequent degradation [88,89]. Most interestingly, an intracellular signal transduction cascade involving p38α (MAPK), GSK-3β and β-catenin has also been reported [90,91] in which p38α (MAPK) inactivates GSK-3β by phosphorylation of its carboxy terminal end, leading to the activation of β-catenin [92]. In the eye, activation of p38α (MAPK) was found to increase permeability of endothelial cells treated with VEGF [90]. In addition, in both primary and SV40-transformed human corneal epithelial cells, expression of the cornified envelope protein small proline-rich protein 1B (SPRR1B), a biomarker for squamous metaplasia, has been shown to be regulated by p38α (MAPK) signaling through a cytokine-induced pathway [93]. Interestingly, adiponectin, a secreted protein produced mainly by adipose tissues that exerts its regulatory influences by binding its corresponding receptors (AdipoR1 and AdipoR2), has been reported to regulate β-catenin signaling in both cementoblasts and hippocampal neural stem/progenitor cells essentially by inactivating the GSK-3β kinase activity, a process that also involves activation of p38α (MAPK) [91,94]. Adiponectin protects various organs and tissues through a yet not fully understood pleiotropic action. It exists in the circulation under three oligomeric complexes: a ~70 kDa low-molecular-weight (LMW) trimer, a ~140 kDa medium-molecular-weight (MMW) hexamer, and a ~300 kDa high molecular weight (HMW) oligomer containing more than 18 monomers [95,96].

To our knowledge, no study ever investigated the expression/secretion of adiponectin and its AdipoR1/R2 receptors in the human cornea. However, a detailed analysis of the hCECs microarray data we accumulated over the years indicated that although the adiponectin transcript was expressed only at very low levels in these cells, they do, however, strongly express both the AdipoR1 and AdipoR2 receptors mRNA transcripts (Appendix A), indicating that they should be capable of responding to alterations in the plasma level of adiponectin. Most interestingly, the murine lacrimal glands were reported to express both adiponectin and adipoR2 mRNA [97] indicating that besides the blood, tears may as well represent a source of secreted adiponectin for hCECs. As a further support for the presence of AdipoR1/R2 in corneal epithelial cells, topically administered adiponectin proved effective to reduce inflammation of the ocular surface in a mouse model of experimental dry eye [98] and corneal neovascularization in a rabbit [99]. It also increased epithelial migration and improved clinical signs and inflammation on the ocular surface after alkali burn, suggesting that adiponectin can promote wound healing in the cornea [100]. A recent proteomic study conducted on the protein cargo of exosomes released by different types of cultured cells provided evidence that HMW adiponectin was present at the surface of the exosome membrane [76]. Although we did not investigate whether our corneal exosomes do transport HMW adiponectin at their surface, such a location would, however, facilitate its interaction with its AdipoR1/R2 receptors and lead either to activation of p38α (MAPK) or to stimulation of endocytosis and cargo delivery through vesicle internalization [101].

HSP27, another mediator that was also increased in hCECs upon addition of exosomes derived from hCECs, has also been linked to cellular proliferation. HSP27 is a small, ubiquitous heat shock protein that belongs to the group of molecular chaperones, which respond to various environmental stresses. HSP27 has been found to be overexpressed in various types of cancers [102,103], where it works as a balance regulator between cell death and survival. In a recent study, HSP27 has been found to promote the epithelial-to-mesenchymal transition and proliferation in colorectal carcinoma cells [104]. HSP27 has also been found to inhibit apoptosis and promote malignant transformation of human bronchial epithelial cells [105]. These results highlight the important role of this mediator in proliferation and survival of epithelial cells. In the wound healing experiment shown on Figure 4A, all exosomes yielded a faster wound closure than the controls. Yet, those isolated from hCECs were distinguished from the others as they clearly yielded a faster closure of scratch-wounded hCECs, whereas exosomes from both hCFs and hCEnCs proved equally efficient. When one looks at the data from the phospho-kinase array (Figure 5B), the only kinase that is activated by all three types of exosomes is HSP27. Interestingly, HSP27 was clearly more activated by hCECs exosomes than by either hCFs or hCEnCs exosomes. As stated above, activation of HSP27 is well known to stimulate cell proliferation and adhesion. Furthermore, it has been shown to be a critical component of the STAT3/HSP27/p38α (MAPK)/Akt survival pathway that also involves phosphorylation of the transcription factor STAT5 [106]. Interestingly, both STAT3 and p38α (MAPK) were identified among the few mediators that also have their phosphorylation level increased in hCECs supplemented with hCECs exosomes (but not with those from hCFs (no activated STAT3), nor hCEnCs, (no activated p38α (MAPK)). Consequently, we believe the faster wound closure observed when hCECs are added exosomes from hCECs might have resulted from the activation of the STAT3/HSP27/p38α (MAPK)/Akt survival pathway and that such activation is disrupted by the lack of some important mediators (STAT3 or p38α (MAPK)) when exosomes from either hCFs or hCEnCs are used.

The fact that none of the 82 genes identified as commonly differentially regulated in hCECs by hCECs-, hCEnCs- and hCFs-exosomes were among those also identified as differentially regulated in hCFs (110 genes) and hCEnCs (228 genes) by the same three types of exosomes is particularly interesting in that it suggests that their respective protein cargo do exert very distinctive, cell-type specific impact on the targeted cell’s transcriptome. This is supported by the recent finding that despite exosomes isolated from various cell types do share common proteins, between 9 and 28% of their protein content is unique to each of them, as could be demonstrated by mass spectrometry and confirmed by immunoblotting [76]. This may explain, at least in part, why they are predicted (by IPA analyses) to differently affect the biological functions examined. From this striking result, we may also hypothesize that the bioactive molecules that the exosomes carry would not be what would have the most significant impact. Rather, what seems most significant is how the cells that receive the exosomes react to them, more particularly which receptors are triggered and which intracellular signaling pathways are activated. Each corneal cell type is indeed very distinct from one another, with striking differences both from an anatomical and functional point of view. Corneal epithelial cells act together to form a barrier that protects the cornea. These cells undergo constant renewal, going through several cycles of proliferation before entering terminal differentiation. Stromal fibroblasts are elongated, highly proliferative cells in vitro that maintain stromal homeostasis by participating in the collagens, glycosaminoglycans and matrix metalloproteinases (MMPs) production. Corneal endothelial cells are polygonal cells that express a high density of sodium-potassium pumps (Na^+^, K^+^-ATPase) at their basolateral membrane. These ion pumps allow the endothelial cells to carry out their main function, i.e., maintaining the state of deturgescence (state of partial dehydration) of the corneal stroma, which is necessary in order to allow optimal light transmission. Unlike corneal epithelial cells, corneal endothelial cells do not proliferate in vivo [107]. Not surprisingly, the three different corneal cell types do not show that same sensitivity and reactivity towards a large number of factors, including growth factors, cytokines and extracellular matrix proteins. For example, high affinity EGF receptors (EGFR) are known to be present on the surface of epithelial and endothelial cells, but absent on stromal keratocytes [108]. Among the other notable differences, the EGF-R, IL-1R, PDGF-R receptors are mainly or exclusively expressed by fibroblasts and the KGFR and c-met receptors, in contrast, are mainly expressed by epithelial cells [109], which may be elevated during wound healing [110]. The integrin expression profile is also significantly different between the three corneal cell types with the heterodimers α6β4 and αvβ6 being exclusively identified in the epithelium and the heterodimer α4β1 in the fibroblasts [111]. All those differences are likely to lead to cell specific signal integration and therefore are likely to contribute to the distinctive, cell-type specific impact that exosomes exert on the targeted cell’s transcriptome.

It is worthy noticing that in a biological context, communication between the three layers of the cornea through exosomes’ release could be restricted in some ways. Indeed, exosomes from hCECs are less likely to reach and impact hCEnCs and vice versa due to the physical distance and the physiological barriers that separate these two layers. Studies in tissue-engineered corneal models and in ex vivo rabbit corneas have demonstrated that EVs can be visualized within the collagen matrix of the stroma and within the corneal endothelium [44]. EVs also appear to penetrate the Descemet’s membrane [44], supporting the idea of a stromal-endothelial cell communication. As for the basement membrane, it appears to limit the diffusion of EVs to the stroma. However, if the BM is disrupted, as it often occurs in the case of corneal epithelial injury, EVs may gain access to the underlying stroma and therefore impact on corneal fibroblasts [41,112]. Consequently, the most relevant results, from a biological point of view, concern the impact of hCECs exosomes on hCFs and vice versa and the impact of hCFs exosomes on hCEnCs and the other way around.

In summary, we demonstrated that the three corneal cell types indeed release EVs that could be effectively enriched by differential ultracentrifugation. Characterization of our samples allows determining that the enriched EVs were positive for exosomal markers CD63, CD9 and CD81 and range in size from 50 to 150 nm in diameter, supporting the idea that our samples are mainly composed of exosomes. In our study, exosomes turned out to be real functional entities, since once internalized, they could stimulate cell proliferation of hCECs. They also demonstrated an exciting potential to enhance corneal epithelial wound healing in a monolayer model. Finally, cornea-derived exosomes also had a cell-type specific impact on the gene expression pattern of hCECs with the differential regulation of genes whose encoded protein products are involved in proliferation, migration and differentiation, three functions that are of great importance in the wound healing process.

## 4. Materials and Methods

This study was conducted in accordance with our institution’s guidelines and the Declaration of Helsinki. The protocols were approved by the CHU de Québec—Université Laval hospital and Université Laval Committees for the Protection of Human Subjects (ethic code: DR-002-955, protocol renewal approved on 21 February 2022).

### 4.1. Cell Isolation and Culture

hCECs, hCFs and hCEnCs were isolated from the cornea of normal human eyes (obtained from the Banque d’Yeux Nationale of the Centre Universitaire d’Ophtalmologie; CHU de Québec, Hôpital du Saint-Sacrement, Québec, QC, Canada). To isolate hCEnCs, the Descemet membrane was carefully peeled off and incubated overnight in culture medium at 37 °C. It was then digested with EDTA 0.02% buffered solution (Sigma-Aldrich, St. Louis, MO, USA) for 1 h. Cells were detached by pipetting up and down with a flamed-polished pipette [113]. hCEnCs were then seeded on fibronectin/collagen (FNC)-coated plastic culture dishes and grown until they reached confluence in a proliferation medium (Opti-MEM-I medium supplemented with 8% fetal bovine serum, 5 ng/mL epidermal growth factor, 0.08% chondroitin sulfate, 20 μg/mL ascorbic acid and 100 IU/mL of penicillin and 100 μg/mL of streptomycin). The culture medium was then replaced with a maturation medium (Opti-MEM-I medium supplemented with 8% fetal bovine serum and penicillin/streptomycin) when hCEnCs reached confluence and were grown further for an additional 7 to 28 days [114]. hCEnCs were used between passages 2 and 6.

To isolate hCECs and hCFs, post-mortem corneas were incubated with dispase (Roche Diagnostics, Laval, QC, Canada) in HEPES buffer (MD Biomedicals, Montreal, QC, Canada) overnight at 4 °C to separate the epithelium from the stroma. The stroma was then cut into small pieces and incubated with collagenase H (Sigma-Aldrich) until the ECM was fully digested by the enzyme. hCECs were treated with trypsin for 15 min at 37 °C to separate the cells from each other. hCECs were grown in DH medium (Dulbecco–Vogt modification of Eagle’s medium with Ham’s F12 in a 3:1 ratio supplemented with 5% FetalClone II serum, 5 μg/mL of insulin, 0.4 μg/mL of hydrocortisone, 10 ng/mL of epidermal growth factor, 0.212 mg/mL of isoproterenol hydrochloride (Sigma-Aldrich, Oakville, ON, Canada), antibiotics (100 IU/mL of penicillin, and 25 μg/mL of gentamycin)) on lethally irradiated human fibroblasts feeder layers (iHFL) until they reached confluence [115]. hCECs were used between passages 1 and 4. hCFs were grown in DME medium (Dulbecco–Vogt modification of Eagle’s medium supplemented with 10% fetal calf serum and antibiotics). hCECs and hCFs were used between passages 2 and 6. All cells were grown under 8% CO_2_ at 37 °C and culture medium was changed every 2 or 3 days.

### 4.2. Exosome Enrichment

Exosomes were isolated by differential ultracentrifugation as previously described [116], with minor modifications. Approximately 48 h before the isolation of exosomes, the culture media of confluent hCFs, hCECs and hCEnCs were changed for culture media with exosome depleted-FBS (serum was depleted of exosomes using ultracentrifugation and filtration). Conditioned media were collected and subsequently centrifuged at 300× *g* for 10 min and 2000× *g* for 20 min to remove cells and large debris. The supernatant was then centrifuged at 21,000× *g* for 1 h 30 min at 4 °C to pellet larger microvesicles. For exosome enrichment, the supernatant was centrifuged at 100,000× *g* overnight at 4 °C. Each exosome pellet was resuspended in HEPES-Buffered Saline (HBS) and kept at 4 °C during the experimental procedure. For long time storage, exosomes were kept at −80 °C.

### 4.3. Quantification of Exosomes

An estimate of the protein concentration of our samples was obtained using a NanoDrop 2000 spectrophotometer (Thermo Fisher Scientific, Mississauga, ON, Canada) at 280 nm.

### 4.4. Electron Microscopy

Exosomes were fixed in 2.5% glutaraldehyde (Canemco, Lakefield, QC, Canada) and processed for transmission electron microscopy (TEM). Exosomes samples were diluted (1/100) and stained with 3% uranyl acetate for 20 s and left to dry overnight. Samples were visualized using a JEOL JEM-1230 (Tokyo, Japan) transmission electron microscope at 80 kV. Prior to observation, the grids were hydrophilized with an X-Cite 120Q UV lamp (Excelitas Technologies, Waltham, MA, USA) for a maximum of 1 h to minimize aggregation.

### 4.5. Dynamic Light Scattering (DLS)

The sizes of the isolated exosomes were determined by Dynamic Light Scattering using NanoBrook Omni from Brookhaven Instruments Corporation (Holtsville, NY, USA). Exosomes’ suspensions (500 μL each) were added to disposable plastic cuvettes (#952010051), air bubbles were carefully removed, and measurements were recorded. The apparatus was set to an angle of 90° at 25 °C. After an equilibrium time of 2 min, 10 measurements of 120 s were performed for each sample and values were reported as Effective Diameter. Analysis of the size distribution was performed using the CONTIN algorithm.

### 4.6. Western Blot Analyses

Western blots were conducted using total protein extracts prepared from exosomes samples isolated from hCFs, hCECs and hCEnCs. Exosomes were lysed with TNG-T lysis buffer (15 nM NaCl, 5 mM Tris-HCl, 1% Glycerol, 0.1% Triton X-100) supplemented with protease inhibitor cocktail (Sigma-Aldrich) and protein concentration was evaluated with the Bradford procedure. Western blots were conducted as described [117] using the following primary antibodies: mouse monoclonal antibodies against CD81 (1:1000; 349502, Biolegend, San Diego, CA, USA), CD9 (1:1000; 312102, Biolegend) and CD63 (1:1000; 353013, Biolegend) and a rabbit polyclonal antibody against Cytochrome-c (1:500; SC-7159 H-104, Santa Cruz Biotechnology, Dallas, TX, USA). A peroxidase-conjugated AffiniPure Goat antibody against either mouse or rabbit IgG (1:2500; 115-036-003, Jackson ImmunoResearch Laboratories, Baltimore, PA, USA) was used as a secondary antibody. Primary antibodies were incubated overnight at 4 °C. The secondary antibody was incubated for 90 min at room temperature. The blots were revealed using ECL Plus Western blotting detection system (Thermo Fisher Scientific Inc.).

### 4.7. Exosomes Uptake

Cellular uptake of exosomes by hCECs and hCFs was assessed by confocal microscopy. Exosome were labeled with DiI fluorescent dye (1,1′-dioctadecyl-3,3,3′,3′-tetramethylindocarbocyanine perchlorate; Thermo Fisher Scientific), a lipophilic membrane stain. Briefly, exosomes’ suspensions (100 μL each) were incubated with DiI dye for 20 min at 37 °C in the dark. Then labeled-exosomes were washed, collected and added to hCECs and hCFs cultures. Cells were incubated with labeled exosomes for 24 h prior to fixation in 4% formaldehyde. Prior to immunodetection, cells were permeabilized with 0.2% Triton X-100 for 10 min. Cell nuclei and actin filaments were counterstained with Hoechst reagent 33258 (1:100; Sigma) and phalloidin-Alexa 488 (1:200, Invitrogen, Carlsbad, CA, USA) respectively. Photos were taken with a confocal microscope (LSM 800; Zeiss, Toronto, ON, Canada).

### 4.8. Ki-67 Immunofluorescences

hCECs were seeded in 24-well plates on coverslip glasses at 2 × 10^4^ cells/well in complete DH medium. At 6 h post-seeding, different types of exosomes (800 μg) or vehicle alone (HBS; negative control) were added to cultures. Cells were incubated for 48 h at 37 °C before they were fixed in 4% formaldehyde. Cells were then permeabilized with 0.2% Triton X-100 for 10 min and incubated with the following primary antibody: mouse monoclonal antibody against Ki-67 (1:200, #556003, BD Biosciences, Franklin Lakes, NJ, USA). Samples were washed with PBS before addition of secondary antibody, peroxidase-conjugated AffiniPure Goat anti-mouse IgG488 (1:400, A11059, Invitrogen). All antibodies were diluted in PBS containing 1% bovine serum albumin. Cell nuclei were counterstained with Hoechst reagent 33258 (1:100; Sigma). Coverslips were mounted on glass slides with mounting medium and kept at 4 °C until observation with an epifluorescence microscope (Zeiss Axio Imager Z2 microscope; Zeiss Canada Ltd.). Samples were photographed with a numeric CCD camera (AxioCam MRm; Zeiss Canada Ltd.). Negligible background was observed for controls (primary antibodies omitted). Number of Ki-67-positive cells was counted and expressed as a percentage of the total number of cells.

### 4.9. Scratch Wound Assay

hCECs (1.86 × 10^5^ cells) were plated with 1.68 × 10^5^ iHFL/cm^2^ in 60 mm petri dish in DH medium. When cells reached confluence, a 1.3 cm-large × 5 cm-long scratch was created in the center of the plate using a policeman (Sarstedt, Nümbrecht, Germany) prior to addition of exosomes from the three corneal cell types (Exos hCFs, Exos hCECs, Exos hCEnCs; 800 μg) or HBS (negative control). Wound closure was monitored on triplicates, and photographs were collected at various time intervals (0, 48, 96 and 120 h). Fresh exosomes were added to the cultures at the time of media changes every 2 days. The wound surface over time was measured using the ImageJ software (version 2.1.0, Wayne Rasband, National Institute of Health (NIH), Bethesda, MD, USA).

### 4.10. Phosphokinase Arrays

The relative levels of 37 different human phosphorylated protein kinases were determined using a membrane-based antibody array (R&D Systems, Minneapolis, MN, USA) according to the manufacturer’s instructions. Briefly, equal amounts (300 µg) of cell lysates prepared from hCECs were incubated overnight with the phosphokinase array membrane. The membrane was then washed to remove unbound proteins and incubated with a mixture of biotinylated detection antibodies. Streptavidin-HRP and chemiluminescent detection reagents were applied, and the signal produced at each captured spot quantified using ImageJ (from Wayne Rasband, NIH). The intensity of each spot including the positive control was quantified and the background was subtracted.

### 4.11. Gene Profiling

All microarray analyses were conducted by the CUO-Recherche gene profiling service (Québec, QC, Canada) as previously described [118,119]. Total RNA was isolated from hCECs, hCFs and hCEnCs exposed or not for 48 h to various populations of exosomes using the RNeasy Mini Kit (QIAGEN, Hilden, Germany) and its quality determined (2100 Bioanalyzer, Agilent Technologies, Santa Clara, CA, USA). Labeling of Cyanine 3-CTP labeled targets, their hybridization on a G4851A SurePrint G3 Human GE 8x60K array slide (Agilent Technologies) and data acquisition and analyses were all performed as previously reported [118]. All data generated from the arrays were analyzed by robust multi-array analysis (RMA) for background correction of the raw values. They were then transformed in Log2 base and quantile normalized before a linear model was fitted to the normalized data to obtain an expression measure for each probe set on each array. Scatter plots and heat maps were generated using the ArrayStar V4.1 (DNASTAR, Madison, WI, USA) software. All microarray data presented in this study comply with the Minimum Information About a Microarray Experiment (MIAME) requirements. (GSE # will be provided prior to publication).

### 4.12. Bioinformatics and Ingenuity Pathway Analyses

The ArrayStar microarray differential linear expression data from hCFs, hCECs and hCEnCs grown either alone or in the presence of exosomes were uploaded to and analyzed with Ingenuity Pathway Analysis (IPA; QIAGEN Inc., https://www.qiagenbioinformatics.com/products/ingenuitypathway-analysis, accessed on 14 March 2022) software in order to compute and visualize causal gene interaction networks around selected cellular functions of interest in hCFs, hCECs and hCEnCs [120,121].

### 4.13. Statistical Analyses

In Figure 3 and Figure 4, Kruskal–Wallis and Mann–Whitney tests were employed to determine statistical significance for comparison of the groups in the Ki-67 immunofluorescence quantification and scratch assays. *p* < 0.05 was considered significant. All data are expressed as mean ± SEM.

## Figures and Tables

**Figure 1 ijms-23-12201-f001:**
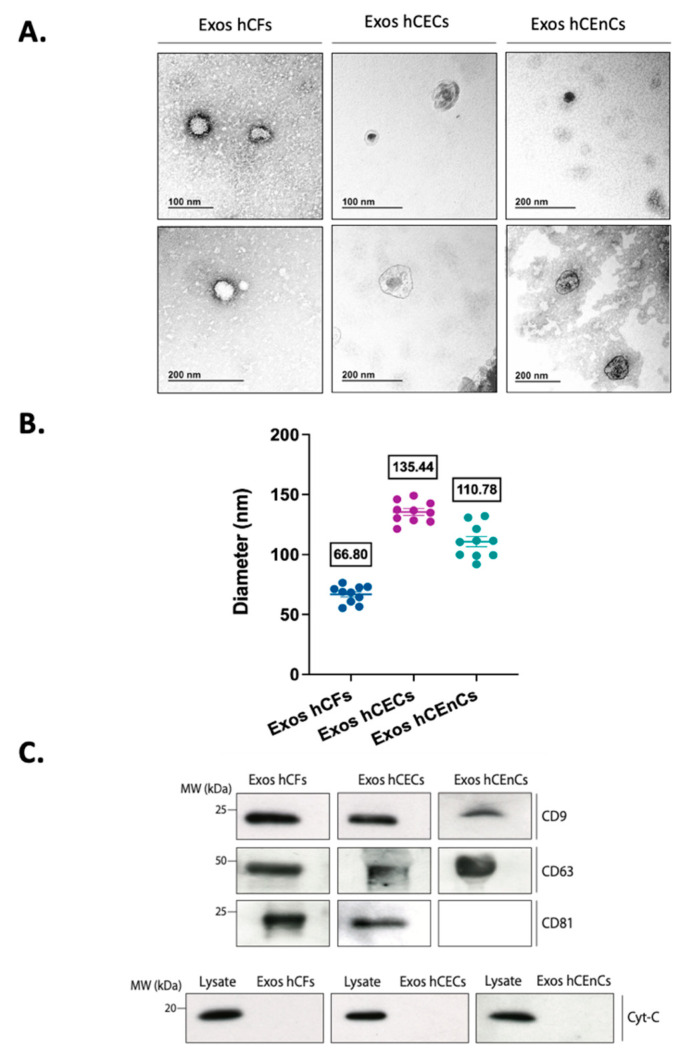
Characteristics of the exosomes isolated from the three corneal cell types. (**A**) TEM images of exosomes isolated from hCFs, hCECs and hCEnCs showing a range of exosomal size from 30–150 nm and a round morphology. Scale bars: 100 nm or 200 nm. (**B**) The diameter of the exosomes isolated from the three corneal cell types were analyzed by DLS. The means for the 10 measurements (repeated measures) were calculated and plotted on a graph. (**C**) Western Blot analysis of three specific exosomal markers (CD9, CD63 and CD81) as well as one negative marker (cytochrome c) in exosomes isolated from conditioned medium of hCFs, hCECs and hCEnCs. Corresponding cell lysate were used as a positive control for cytochrome c.

**Figure 2 ijms-23-12201-f002:**
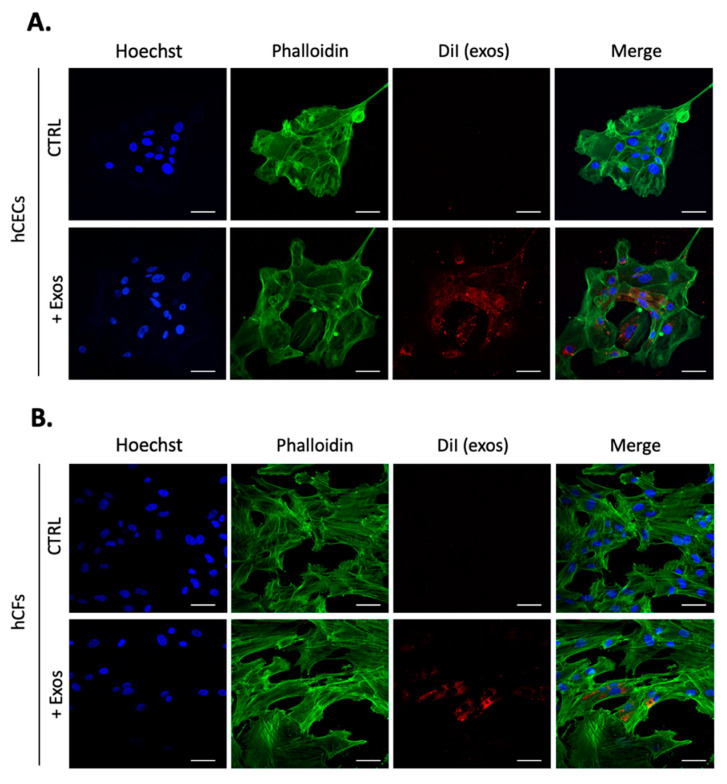
Exosomes uptake in hCECs and hCFs. Exosomes were labeled with the lipophilic dye, DiI. DiI-labeled exosomes were incubated (direct addition to the culture medium) with either hCECs or hCFs for 24 h prior to fixation and immunofluorescence staining. HBS, which was used to resuspend exosomes, was used as control (CTRL). Cells were stained with phalloidin, which exhibits green fluorescence, diI-labeled exosomes appeared in red and nuclei (Hoechst staining of DNA) in blue. Panels (**A**,**B**) show representative images of diI-labeled exosomes uptake by hCECs and hCFs, respectively. Scale bars: 50 μm.

**Figure 3 ijms-23-12201-f003:**
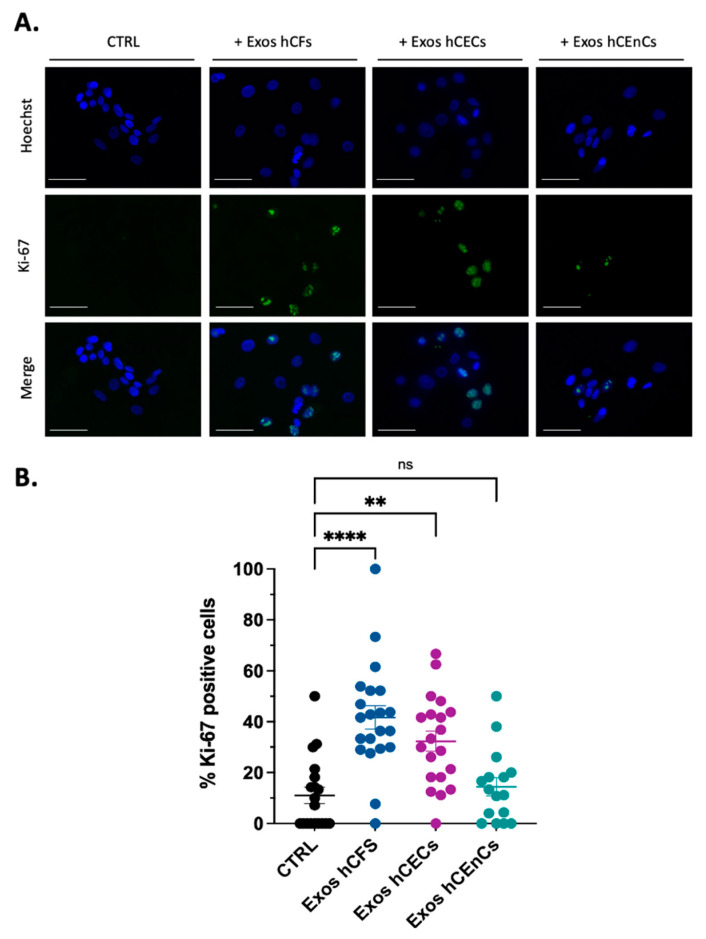
Effect of exosomes from the three corneal cell types on hCECs proliferation. hCECs were incubated with 800–1000 μg exosomes isolated from medium of hCFs, hCECs and hCEnCs conditioned for 48 h. HBS was used as control (CTRL). Cells were fixed and Ki-67 was labeled by indirect immunofluorescence. Panel (**A**) shows representative images of Ki-67 expression in hCECs. Nuclei were counterstained with Hoechst 33258 reagent (blue). Scale bars: 100 μm. (**B**) The number of Ki-67 positive cells was calculated and plotted on graph. Between four and eight photos per condition were used for the counts. The data is expressed as the mean ± SEM from three independent experiments on two different populations. ** *p* < 0.01, **** *p* < 0.001, ns: not significant.

**Figure 4 ijms-23-12201-f004:**
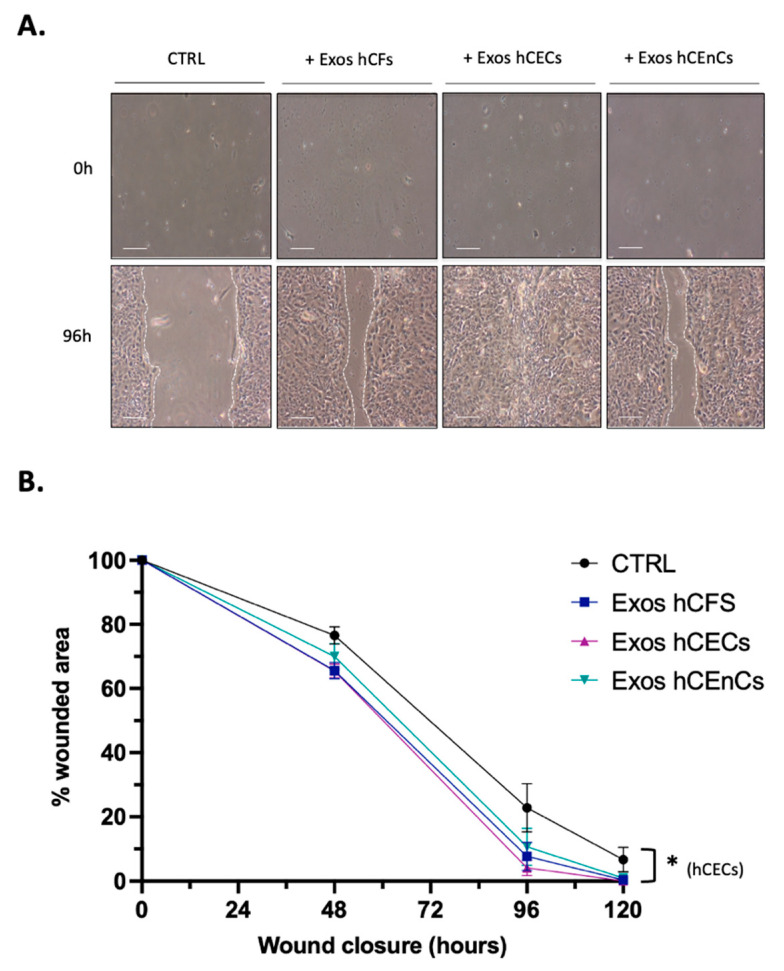
Impact of the three types of exosomes on wound closure of hCECs. (**A**) hCECs were grown as a monolayer and scratch-wounded. 800–1000 μg exosomes from each corneal cell type (hCFs, hCECs and hCEnCs) were separately administrated every 48 h and hCECs were allowed to recover. Scratches were photographed 0, 48, 96 and 120 h after wounding to monitor the healing process. As a negative control, hCECs were incubated with the vehicle alone (HBS). Scale bars: 200 μm. (**B**) The means of the wound surfaces remaining for each condition were calculated for each time interval and plotted on graph. Data is expressed as the mean ± SEM. For each time interval, results were compared to controls and the difference was considered statistically significant when * *p* < 0.05.

**Figure 5 ijms-23-12201-f005:**
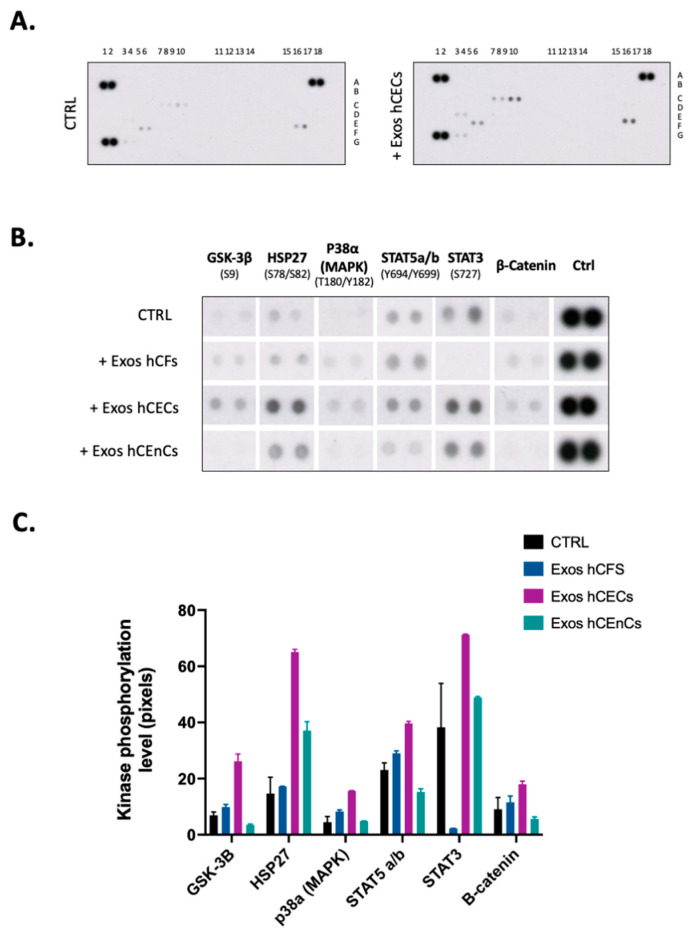
Impact of the exosomes isolated from the different corneal cell types on the phosphorylation of signal transduction mediators in hCECs. (**A**) Cells were cultured 48 h with hCFs-, hCECs-, hCEnCs-exosomes or the vehicle (HBS). Cell lysates prepared from hCECs were used as sources of proteins for the phosphokinase assays conducted as detailed in Section 4. The results obtained for hCECs exposed solely to the vehicle (HBS) or to hCECs exosomes are shown as examples. (**B**) Duplicate spots corresponding to the kinases (GSK-3β, p38α (MAPK), STAT5a/b, STAT3, β-catenin) whose phosphorylation is the most altered in monolayer-cultured hCECs after the addition of exosomes from hCECs, HCFs or hCEnCs. hCECs exposed only to the vehicle were used as negative control (CTRL). (**C**) The phosphorylation level (in pixels) for each spot was determined with the ImageJ Software and plotted on graph. One of two representative experiments conducted using two distinct populations of hCECs is shown.

**Figure 6 ijms-23-12201-f006:**
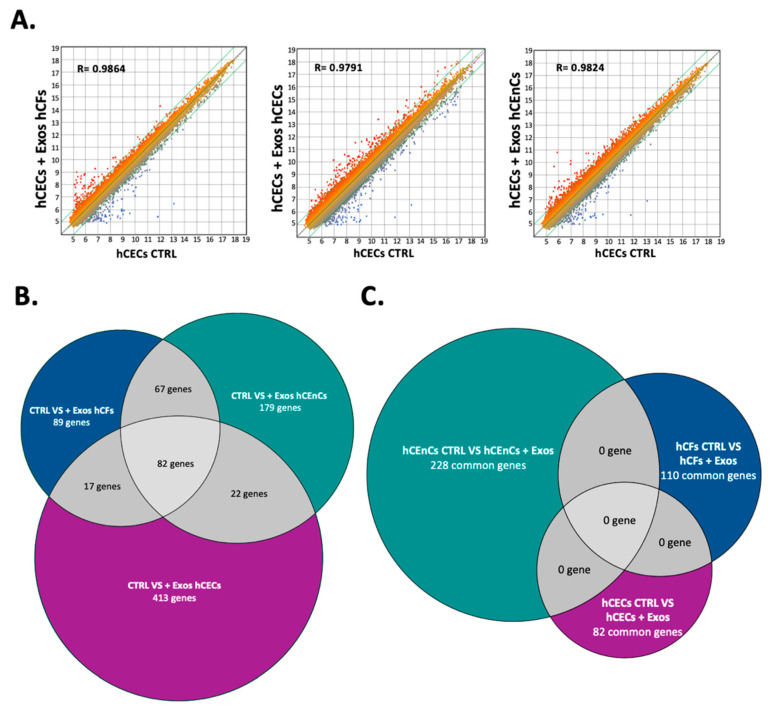
Microarray analysis of the gene expression pattern of hCECs cultured in the presence of hCFs-, hCECs- or hCEnC-exosomes. (**A**) Scatter plot of log2 of signal intensity from 60,000 different targets covering the entire human transcriptome of hCECs + Exos hCFs (first graph), hCECs + Exos hCECs (second graph) or hCECs + Exos hCEnCs (third graph) in the y-axis as a function of hCECs CTRL (no added exosomes) in the x-axis. (**B**) Analysis of commonly regulated genes in hCECs. Venn diagram depicting the number of genes differently regulated by at least a two-fold factor between hCECs CTRL and hCECs + Exos hCFs (upper left; 89 genes); hCECs CTRL and hCECs + Exos hCECs (bottom; 413 genes) and those specific to the hCECs + Exos hCEnCs condition (upper right; 179 genes). Differently regulated genes shared between two groups are indicated at the intersections (17, 22 and 67 genes) and differently regulated genes common for three groups are indicated in the middle (82 genes). (**C**) Analysis of differentially regulated genes that are common following exposure to exosomes from each of the three different corneal cell types. In this Venn diagram, each circle indicated the number of genes differentially regulated by at least two-fold in hCFs, hCECs and hCEnCs exposed to exosomes from the three different sources. As shown in the panel (**B**), 82 genes are commonly differently regulated in hCECs CTRL vs. hCECs + Exos (from any source). As for hCEnCs and hCFs, there are respectively, 228 genes (upper left) and 110 genes (upper right) that are commonly differently regulated. All these genes are unique to a given cell population, none of those genes are shared between the three corneal cell types (middle; 0 gene).

**Figure 7 ijms-23-12201-f007:**
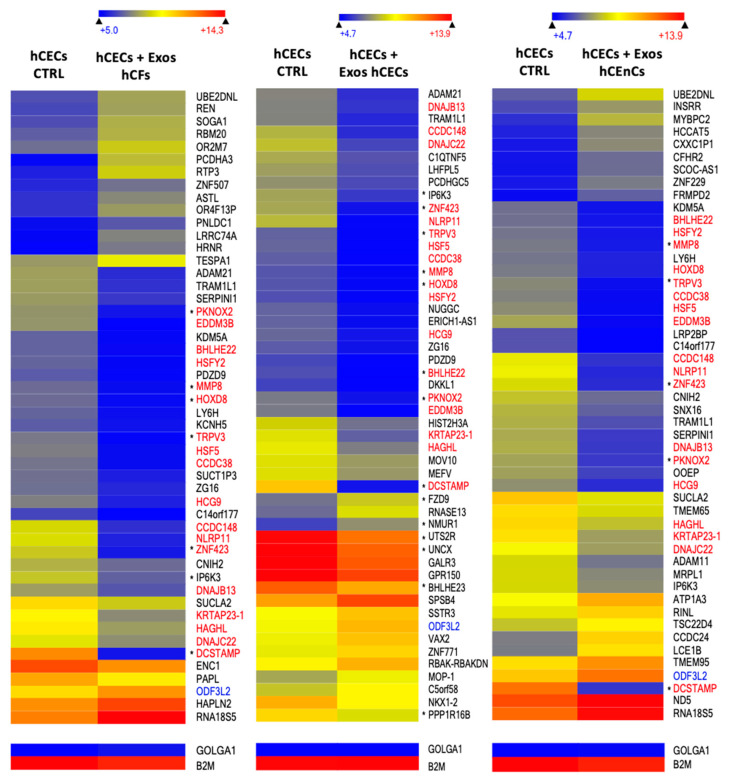
Gene expression pattern of hCECs cultured in the presence of hCFs-, hCECs- or hCEnC-exosomes. Heatmap representation of the 50 most differentially regulated genes in hCECs CTRL (no added exosomes) against hCECs + Exos hCFs (first heatmap), hCECs + Exos hCECs (second heatmap) or hCECs + Exos hCEnCs (third heatmap). Gene names indicated in red correspond to genes whose transcription are commonly downregulated in all three conditions whereas those in blue are upregulated. An asterisk placed before the gene name indicates that this gene has been associated with at least one function of interest in the IPA analysis. Microarray data for the golgin subfamily A member 1 (GOLGA1) and β2-microglobulin (B2M) housekeeping genes that are expressed, respectively, at low and very high levels in all cell types are also shown.

**Figure 8 ijms-23-12201-f008:**
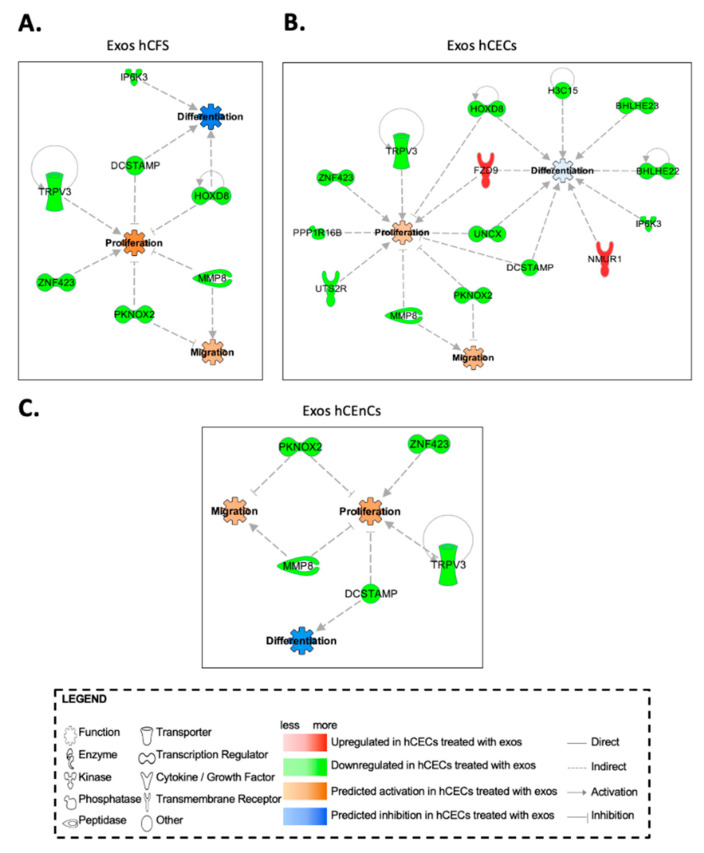
IPA generated interactome derived from differentially expressed genes in hCECs when supplemented with hCFs—(**A**), hCECs—(**B**) and hCEnCs—(**C**) exosomes. Computationally predicted biological functions of interest (proliferation, migration and differentiation) are identified with bold labels and colored either orange or blue depending on whether they are predicted to be activated or inhibited respectively in cultures supplemented with exosomes. Differentially expressed genes present in our datasets are labeled with non-bold text and are colored either green or red depending on whether they were up- or downregulated respectively in cultures supplemented with exosomes. Lines indicate gene—gene and gene—function relationships (full lines for direct relationships and dotted lines for indirect ones) based on IPA’s database. Functions are indicated in bold.

## Data Availability

All microarray data presented in this study can be accessed at NCBI Gene Expression Omnibus (GEO# GSE215102; https://www.ncbi.nlm.nih.gov/geo/query/acc.cgi?acc=GSE215102, accessed on 9 October 2022).

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
