# Peer review of "Impact of Exosomes Released by Different Corneal Cell Types on the Wound Healing Properties of Human Corneal Epithelial Cells"

_ijms, 2022, doi:10.3390/ijms232012201_

Round 1

Reviewer 1 Report

The paper concerns an important topic, is well written and presents interesting data. There are, however, a few comments: 

The observation “Alexa Fluor 488-conjugated phalloidin which selectively label F-actin revealed that exosomes appear to be preferentially located at the periphery of the nuclei” (lines 156-158) should be better illustrated as that is not quite clear for hCEC (Fig.2A), while somewhat better illustrated for hCFs (Fig.2B). Also, please elaborate on why such data were not presented for hCEnC, what was the purpose of F-actin localization and what is the significance of this observation.

The statement (line 515) can be expanded by adding more recent references, e.g.: The KGFR and c-met receptors, in contrast, are mainly expressed by epithelial cells [109], which may be elevated during wound healing (Wilson SE, Chen L, Mohan RR, Liang Q, Liu J. Expression of HGF, KGF, EGF and receptor messenger RNAs following corneal epithelial wounding. Exp Eye Res.1999;68(4):377-97).

The authors should probably avoid using the term “signalization” which is not quite conventional. 

Author Response

R1.1. The observation “Alexa Fluor 488-conjugated phalloidin which selectively label F-actin revealed that exosomes appear to be preferentially located at the periphery of the nuclei” (lines 156-158) should be better illustrated as that is not quite clear for hCEC (Fig.2A), while somewhat better illustrated for hCFs (Fig.2B). Also, please elaborate on why such data were not presented for hCEnC, what was the purpose of F-actin localization and what is the significance of this observation.

It is true that such a location is not quite obvious. We can see a few cells for which it is the case, but not for all of them. We therefore replaced “preferentially located at the periphery of the nuclei” by  “located through the cytoplasm and occasionally concentrated around the nuclei” in the text to avoid any confusion. As we mainly focus on the impact of exosomes on hCECs, we thought unnecessary to assess the uptake of exosomes in hCEnCs. Uptake in hCFs was conducted as a comparative analysis essentially because hCFs were readily available and fitted for the design of the experiment, which was not the case for hCEnCs that are much more difficult to obtain and need a long maturation time in culture. Moreover, for this experiment, we used exosomes derived from hCECs. Therefore, considering the physiological context, it was more relevant to evaluate the uptake of these exosomes on the nearby cells, namely hCECs and hCFs. Finally, F-actin was used only to delineate cells.

R1.2. The statement (line 515) can be expanded by adding more recent references, e.g.: The KGFR and c-met receptors, in contrast, are mainly expressed by epithelial cells [109], which may be elevated during wound healing (Wilson SE, Chen L, Mohan RR, Liang Q, Liu J. Expression of HGF, KGF, EGF and receptor messenger RNAs following corneal epithelial wounding. Exp Eye Res.1999;68(4):377-97).

We have now added the reference suggested by the reviewer to the Discussion section of the revised manuscript.

R1.3. The authors should probably avoid using the term “signalization” which is not quite conventional

This has been corrected in the text of the revised manuscript, as requested.

Reviewer 2 Report

In this manuscript, the authors showed that exosomes from all three corneal cell types facilitate wound healing. They provided further evidences showing that several kinase pathways were modified following uptake of exosomes. Finally, transcriptomic analysis showed a cell-type specific response to the exposure of exosomes.

Here are a few concerns for the authors.

1. In addition to no exosome control, it is recommended to also include exosomes from an irrelevant cell type (e.g. HEK293) to properly control the potential impurities from the purification procedures.

2. How did the authors determine the amount of exosomes used in the cell uptake experiment? Did the authors normalize the exosomes with protein content? Or based on the exosome particle numbers applied to each cells?

3. Based on the data, in the phospho-kinase proteome profiler assay, hCECs respond differently to exosomes from different cell types, could the authors provide more explanation/discussion on the reason why they observed very similar effect from different exosomes in the wound healing experiment? 

Author Response

R2.1. In addition to no exosome control, it is recommended to also include exosomes from an irrelevant cell type (e.g. HEK293) to properly control the potential impurities from the purification procedures.

Exosomes from HEK293 could have been used as a control to verify that the observed effect was indeed specific to human corneal exosomes. However, their impact could have been difficult to explain and would not have made it possible to control for potential impurities. Indeed, using the same purification procedure, the same impurities would have been found in the samples. The use of another purification method could have been employed in order to minimize the impurities, but this would however have introduced significant heterogeneity into the study. Besides, a procedure we used to demonstrate that EVs used in the present study were not contaminated with cellular debris was to demonstrate the absence of cellular proteins that are known not to be present in EVs, such as cytochrome C (Figure 1C).

R2.2. How did the authors determine the amount of exosomes used in the cell uptake experiment? Did the authors normalize the exosomes with protein content? Or based on the exosome particle numbers applied to each cells?

Each exosome sample was normalized by measuring is protein content. However, for this specific experiment, which was a qualitative rather than a quantitative analysis, we used a volume (100 ml) of exosomes rather than a quantity (as for the proliferation and scratch assays).

R2.3. Based on the data, in the phospho-kinase proteome profiler assay, hCECs respond differently to exosomes from different cell types, could the authors provide more explanation/discussion on the reason why they observed very similar effect from different exosomes in the wound healing experiment? 

In the wound healing experiment shown on Fig. 4A, although all exosomes yielded a faster wound closure than the controls, yet exosomes isolated from hCECs distinguished from the others as they clearly yielded a faster closure of scratch-wounded hCECs, whereas exosomes from both hCFs and hCEnCs were equally efficient. Now, if one looks at the data from the phospho-kinase array (Fig. 5B), the only kinase that is activated by all three types of exosomes is HSP27. Interestingly, HSP27 is clearly more activated by hCECs exosomes than by either hCFs and hCEnCs exosomes. Activation of HSP27 is well known to induce cell proliferation and adhesion. Furthermore, it has been shown to be a critical component of the STAT3/HSP27/p38a (MAPK)/Akt survival pathway that also involves phosphorylation of the transcription factor STAT5 (Sims et al., 2013, PlosOne 8(1):e55509). Interestingly, both STAT3 and p38a (MAPK) were identified among the few mediators that have their phosphorylation level increased in hCECs supplemented with hCECs exosomes (but not with those from hCFs (no activated STAT3), nor hCEnCs (no activated p38a (MAPK)). Consequently, we believe the faster wound closure observed when hCECs are added exosomes from hCECs might have resulted from the activation of the STAT3/HSP27/p38a (MAPK)/Akt survival pathway and that such activation is disrupted by the lack of some important mediators (STAT3 or p38a (MAPK)) when exosomes from either hCFs or hCEnCs are used. These new details have now been added to the Discussion section of the revised manuscript.